# Encephalopathy Caused by Human Parvovirus B19 Genotype 1 Associated with *Haemophilus influenzae* Meningitis in a Newborn

Noely Evangelista Ferreira [1,2,†], Antonio C. da Costa [1,2,†], Esper G. Kallas [1], Cassia G. T. Silveira [3], Ana Carolina S. de Oliveira [3], Layla Honorato [1,2], Heuder G. O. Paião [1,2], Silvia H. Lima [1,2], Dewton de M. Vasconcelos [2], Marina F. Côrtes [1,2], Silvia F. Costa [1,2], Tania R. T. Mendoza [1,2], Hélio R. Gomes [4], Steven S. Witkin [2,5] and Maria C. Mendes-Correa [1,2,*]

1   Departamento de Molestias Infecciosas e Parasitarias, Faculdade de Medicina, Universidade de São Paulo, São Paulo 05403-000, Brazil; noely.ferreira@gmail.com (N.E.F.); charlysbr@yahoo.com.br (A.C.d.C.); esper.kallas@usp.br (E.G.K.); layla.honorato@usp.br (L.H.); heuder.paiao@gmail.com (H.G.O.P.); silvialima.lima@usp.br (S.H.L.); marinafarrel23@gmail.com (M.F.C.); silviacosta@usp.br (S.F.C.); tozetto@usp.br (T.R.T.M.)
2   Instituto de Medicina Tropical, Faculdade de Medicina, Universidade de São Paulo, São Paulo 05403-000, Brazil; dmvascon@usp.br (D.d.M.V.); switkin@med.cornell.edu (S.S.W.)
3   Division of Clinical Immunology and Allergy, School of Medicine, University of São Paulo, São Paulo 05403-000, Brazil; terrassani@gmail.com (C.G.T.S.); soares.acso@gmail.com (A.C.S.d.O.)
4   Laboratório de Investigação Médica LIM 15, Hospital da Clinicas da, Faculdade de Medicina da, Universidade de São Paulo, São Paulo 01246-903, Brazil; helio.gomes@hc.fm.usp.br
5   Department of Obstetrics and Gynecology, Weill Cornell Medicine, 1300 York Avenue, New York, NY 10065, USA
*   Correspondence: cassiamc@uol.com.br
†   These authors contributed equally to this work.

**Abstract:** Parvovirus B19 infection is associated with a wide range of clinical manifestations, from asymptomatic to severe neurological disorders. Its major clinical symptoms, fever and rash, are common to multiple viruses, and laboratory tests to detect B19 are frequently not available. Thus, the impact of B19 on public health remains unclear. We report the case of a 38-day old girl admitted to São Paulo Clinical Hospital, Brazil, with an initial diagnosis of bacterial meningitis, seizures, and acute hydrocephalus. Antibiotic therapy was maintained for one week after admission and discontinued after negative laboratory results were obtained. Nine days after symptoms onset, a cerebral spinal fluid (CSF) sample revealed persistent pleocytosis. The complete B19 complete genome was subsequently identified in her CSF by a metagenomic next-generation sequencing approach. This report highlights the possible involvement of B19 in the occurrence of acute neurological manifestations and emphasizes that its possible involvement might be better revealed by the use of metagenomic technology to detect viral agents in clinical situations of unknown or uncertain etiology.

**Keywords:** human parvovirus B19; encephalopathy; metagenomic next-generation sequencing

## 1. Introduction

The human parvovirus B19 (B19) is a linear, non-enveloped DNA virus, classified in genus Erythroparvovirus of the Parvoviridae family [1] with three distinct genotypes: 1, 2, and 3 [2]. It is a highly infectious virus, disseminated by respiratory secretions, transfusion of infected blood products, and vertically from mother to child [3–5]. Most cases of B19 infection are asymptomatic, occur more frequently in childhood, and are not uniform across the population because seroprevalence ranges from 40% to 87% [6–8].

B19 is associated with a wide range of clinical manifestations, from asymptomatic forms to severe neurological disorders [8–10]. Encephalitis and encephalopathy are the



most common consequences of parvovirus B19 infection in the central nervous system (CNS) and account for 39% of the total B19-associated neurological sequelae [11].

In Brazil, neurological consequences of a parvovirus B19 infection are considered rare, but laboratory tests are not always available to rule out their occurrence [12–14].

Metagenomic next-generation sequencing (mNGS) technology is a high-throughput approach that can simultaneously investigate all genetic material in a biologic sample. It has been utilized as a very useful exploratory method for the identification of viral infections of undetermined etiology when specific agents are not detected by standard diagnostic tests [15–18].

Although B19 has previously been reported to cause a variety of clinical symptoms in patients with different immune disorders, B19 infection in patients younger than one year of age is not common. Here, we report the use of mNGS for the diagnosis of an unusual case of acute viral meningitis in a one-month-old newborn, admitted to the Hospital das Clinicas da Faculdade de Medicina, Sao Paulo University São Paulo, Brazil, with fever, *Haemophilus influenzae* meningitis, and hydrocephalus secondary to the infection. Using metagenomics as an alternative method to aid in the diagnosis of meningitis, we detected and characterized the entire genome of B19 from the cerebrospinal fluid (CSF) of this newborn.

## 2. Setting and Case Report

The present case report evolved from an analysis of CSF samples collected from patients with suspected CNS infection enrolled in a hospital-based surveillance study conducted in Sao Paulo, Brazil, from March 2018 to November 2019. A total of 600 CSF samples were collected at the Hospital das Clinicas da Faculdade de Medicina da Universidade de Sao Paulo, a public hospital (300 samples) and from other different private hospitals in the same city (300 samples), from individuals (both genders and any age group) with clinical suspicion of acute infectious encephalitis or meningitis.

After collection, as per the study protocol, all CSF samples were tested for a range of pathogens using the diagnostic work-up performed in the clinical study which initially included the latex agglutination test; India ink staining; and bacterial, fungi, and tuberculosis cultures. The metagenomic approach was performed as a complementary exploratory analysis to determine the etiology of cases of meningitis, encephalitis, or acute meningoencephalitis in which no cause was identified after this previous screening. A specific infectious etiology was defined in 300 cases but remained unknown in the remaining 300 cases.

The remaining volume of the CSF samples were stored at −80 °C for further testing in the Virology Laboratory at the Tropical Medicine Institute at Sao Paulo University Medical School where a molecular screening by real time PCR for Human Enterovirus, Parvovirus B19, Parechovirus [19,20], and Herpesvirus was performed in a viral panel as with some modifications [21,22]. Among all 600 samples tested, only one sample (0.17%) was positive for Parvovirus B19 by real time PCR.

*Case Report*

A 38-day old girl was admitted to the Hospital das Clinicas da Faculdade de Medicina da Universidade de Sao Paulo with a diagnosis of bacterial meningitis, seizures, and acute hydrocephalus on 28 May 2019. She had been delivered by cesarean section without complications and achieved normal growth until the 30th day of life, when she presented with fever and vomiting. She was immediately referred for additional analysis. In the hospital unit, meningitis due to *Haemophilus influenzae* was diagnosed by a polymerase chain reaction (PCR) assay, and antibiotic therapy was initiated (Ceftriaxone, 50 mg/kg/dose, to a maximum of 2 g, 24 hourly). Her symptoms worsened to persistent seizures requiring orotracheal intubation (OTI) and sedation. Computer tomography (CT) performed 2 days after symptom onset revealed supratentorial hydrocephalus, in addition to diffuse cerebral edema. The baby was then referred to our hospital for hydrocephalus management, where

she was admitted to the neonatal intensive care unit on 28 May 2019. Blood, urine, and CSF samples were collected to confirm the diagnosis of bacterial meningitis and to rule out other co-morbidities.

Routine analyses and cultures of CSF, blood, throat swabs and urine yielded no pathogenic growth. Antibiotic therapy was maintained for an additional one week after admission and was discontinued after negative laboratory tests were obtained. On admission to our hospital (nine days after symptoms onset) a CSF sample indicated pleocytosis (217 leucocytes/mL, with 18% lymphocytes and 67% neutrophils, monocytes 10%, plasma cells 2%, eosinophils 2%, basophils 1%), along with 1 red blood cell/mL, protein content 247 mg/ dL (60–80 mg/dL), lactate level 36.9 mg/dL (10.0 to 22.0 mg/dL) and glucose concentration 40 mg/dL (2/3 glycemia rate).

During her hospitalization (May 2019 to June 2019), other CSF samples were collected, revealing persistence of cytological changes, particularly an increase in the number of red blood cells. These alterations persisted until 40 days after the onset of symptoms, in the absence of identification of any etiological agent. The patient's initial CSF sample was subsequently subjected to a more complete etiological investigation.

Computer tomography performed 10 days after the onset of symptoms showed supratentorial hydrocephalus, diffuse cerebral edema, diffuse subarachnoid hemorrhage, and multiple hypodense areas affecting the cortex and white matter of the frontal, temporal, parietal, and occipital lobes. An external ventricular drain (EVD) was inserted, and the patient remained hospitalized for three months for seizure control. She was discharged after stabilization of her general condition. During the clinical evolution, there was no skin rash described by the attending physicians or the patient's relatives. A serological test for B19 was not performed.

Three years after the initial diagnosis of bacterial meningitis, the child is still being followed up at our hospital, with neurologic sequelae and a diagnosis of cerebral palsy.

## 3. Materials and Methods

*Assessment of Sample*

Briefly, 500 μL of the CSF was centrifuged at 12,000× *g* for 10 min, and total nucleic acids were extracted from 200 μL of the supernatant using the Extracta 96 Fast Kit (Loccus São Paulo, SP, Brazil). Afterwards, cDNA synthesis was performed with Superscript IV Reverse transcription according to the manufacturer's protocol (Thermo Fisher Scientific Inc., Waltham, MA, USA). A second strand of cDNA was obtained using DNA Polymerase I Large (Klenow) Fragment (Promega Corporation, Madison, WI, USA) and viral nucleic acid dosage was determined according to Quantus (Promega Corporation, Madison, WI, USA). Subsequently, the sample was submitted to a Nextera XT Sample Preparation Kit (Illumina, San Diego, CA, USA) to construct a DNA library, identified by means of double barcodes. For size range selection, Pippin Prep (Sage Science Inc., Beverly, MA, USA) was used to select a 400 bp insert (range 400-700 bp). The library was deep sequenced using the Nova Seq 6000 Sequencer (Illumina, San Diego, CA, USA) with 250 bp ends.

The bioinformatic analysis was performed according to the protocol described by Deng et al. [23]. That shared a percent nucleotide identity of 95% or more were assembled from obtained sequence reads by reassembly. The complete B19 genome was mapped with Geneious Prime® 2022.0.2 software (www.geneious.com, accessed on 5 April 2023). This analysis involved 70 nucleotide sequences.

The classification of the Parvovirus B19 genotypes (G1, G2, and G3) was based on the genetic variability of the nucleotide sequences with a variation between 5% and 13% [2,24].

## 4. Results

*4.1. Laboratory and Clinical Epidemiological Characterization*

The CSF sample was negative for all other etiological agents tested as described above by the latex agglutination test, India ink staining, bacterial, fungi, and tuberculosis cultures and real time PCR for Human Enterovirus, Parechovirus, and Herpesvirus. Parvovirus

B19 was detected with 3.4 log10 copies/mL by real time PCR. Tables 1 and 2 and present a summary of characteristics of the case and demographic data.

**Table 1.** Evolution of CSF analyses throughout case report.

| Collection Date | 28 May | 29 May | 22 June | 6 June | 8 June | 12 June | 16 June | 18 June | 24 June | |
|---|---|---|---|---|---|---|---|---|---|---|
| Collect CSF | CSF1 Admission | CSF2 | CSF3 | CSF4 Case | CSF5 | CSF6 | CSF7 | CSF8 | CSF9 | Reference |
| Cells (mm$^3$) | 217 | 43 | 395 | 213 | 80 | 170 | 72 | 47 | 41 | Until 4 cells |
| Red cells (mm$^3$) | 1 | 906 | 221 | 225 | 118 | 1 | 2 | 0 | 1360 | 0 Red Cells |
| Lymphocytes (%) | 18 | 56 | 5 | 19 | 33 | 28 | 47 | 28 | 35 | 70–80 |
| Monocytes (%) | 10 | 22 | 31 | 47 | 40 | 15 | 18 | 12 | 7 | 60–80 |
| Plasmocytes (%) | 2 | 1 | 0 | 3 | 7 | 37 | 32 | 58 | 51 | N/A |
| Neutrophils (%) | 67 | 17 | 52 | 21 | 14 | 17 | 1 | 2 | 5 | N/A |
| Eosinophils (%) | 2 | * | 2 | 3 | 1 | 2 | 1 | * | 1 | N/A |
| Basophils (%) | 1 | 3 | 0 | 5 | 4 | 1 | 1 | * | 1 | N/A |
| Macrophages (%) | * | 1 | 10 | 1 | 1 | * | 1 | * | * | N/A |
| Total protein (mg/dL) | 247 | 188 | 384 | 301 | 214 | 114 | 273 | 228 | 93 | Lumbar–until 40 |
| Glucose (mg/dL) | 40 | 27 | 21 | 21 | 31 | 30 | 31 | 34 | 55 | 50–70 |
| Lactate (mg/dL) | 36.9 | 31.5 | 36.9 | 36 | 28.8 | 22.5 | 29.7 | 25.2 | 22.5 | 10.0–22.0 |
| Microbiological tests | Negative | Negative | Negative | Negative | Negative | Negative | Negative | Negative | Negative | N/A |
| PCR tests other viruses[a] | * | * | * | Negative | * | * | * | * | * | N/A |
| qPCR Parvovirus B19 | * | * | * | Positive | * | * | * | * | * | N/A |

Abbreviations: * = not tested; N/A = not applicable; CSF = cerebrospinal fluid.

**Table 2.** Evolution of blood analyses throughout case report.

| Date Blood Collection | 28 May | 29 May | 30 May | 3 June | 5 June | Reference Value |
|---|---|---|---|---|---|---|
| | Admission | | | | Case | |
| Erythrocytes ($\times 10^6$/L) | 3.99 | 3.88 | 3.65 | 3.38 | 3.56 | 3.2–4.9 |
| Hemoglobin (g/dL) | 11 | 10.8 | 10.3 | 9.6 | 9.9 | 10.3–13.7 |
| Hematocrit (%) | 32.2 | 32.1 | 30.4 | 28.1 | 29.7 | 34–48 |
| MCV (fL) | 80.7 | 82.7 | 83.3 | 83.1 | 83.4 | 80–114 |
| HCM (pg) | 27.6 | 27.8 | 28.2 | 28.4 | 27.8 | 24–34 |
| MCHC (g/dL) | 34.2 | 33.6 | 33.9 | 34.2 | 33.3 | 28–32 |
| RDW (%) | 17.0 | 17.4 | 17.3 | 16.9 | 16.7 | 11.6–14.8 |
| Leukocytes ($\times 10^3$/mm$^3$) | 22.33 | * | 13.83 | 16.84 | 13.84 | 6–18 |
| Rod neutrophils (%) | 8 | 4 | 2 | * | * | 3.8–4.4 |
| Segmented neutrophils (%) | 70 | 65 | 75 | 38 | 46 | 28–30 |
| Lymphocytes (%) | 16 | 17 | 10 | 32 | * | 57–61 |
| Monocytes (%) | 6 | 5 | 5 | 12 | 4 | 4.8–5.9 |
| Platelets (K/L) | 487 | 466 | 108 | 814 | 1212 | 150–450 |

Abbreviations: * = not tested; MCV = mean corpuscular volume; MCH = mean corpuscular hemoglobin; MCHC = mean corpuscular hemoglobin concentration; RDW = red cell distribution width.

*4.2. Metagenomic Analysis of the CSF Samples*

The Parvovirus B19 complete genome was recovered from the CSF sample by a metagenomic approach. Genotype 1 found in our study had a maximum divergence of 2%. The complete genome sequence (sample ID 63660—Mapped reads 546.585, genome length 5427, coverage 34.214% B19 genome 100) has been deposited in GenBank under the accession number OR200802.

*4.3. Phylogenetic Analysis*

Phylogenetic analyses demonstrated similarity with genotype 1 B19 sequences compared to GenBank sequences. The sequences were aligned using the MAFFT tool [25]. The tree was constructed using the maximum likelihood approach, and branching support was estimated using an ultrafast bootstrap test with 1000 replications using the IQ-Tree tool (http://iqtree.cibiv.univie.ac.at/, accessed on 5 April 2023). Tree was visualized and edited using Figtree version 1.4.2 (http://tree.bio.ed.ac.uk/software/figtree, accessed on 5 April 2023).

The phylogenetic tree of B19 was constructed using a dataset consisting of one sequence generated in this study, along with 69 reference sequences of B19V obtained from GenBank (Figure 1). The maximum likelihood approach, with the GTR plus gamma correction model, was employed for tree construction. Branch support was assessed using

the ultrafast bootstrap method with 1000 replications. Sequence generated in our study are highlighted in red (OR200802). Branches in the tree were color-coded according to the bootstrap values, as shown in the upper left panel of Figure 1. The horizontal bar represents the scale of nucleotide substitutions per site. Genotypes 1 and 2 were collapsed for best visualization of tree. The MEGA11 program was used to align the nucleotide sequences [26].

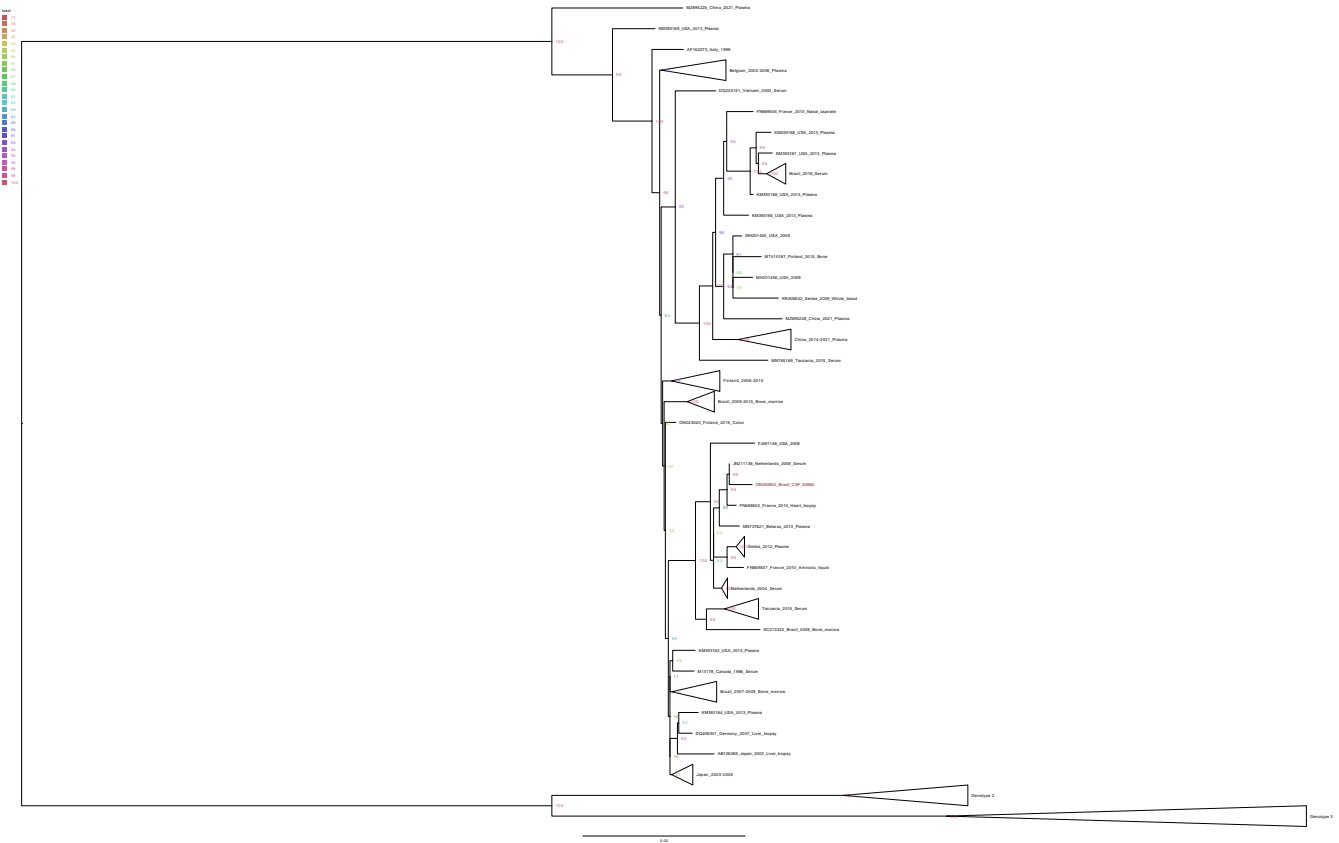

**Figure 1.** Phylogenetic tree of Human Parvovirus B19V.

## 5. Discussion

Parvovirus B19 is common worldwide and in addition to its classic manifestations, has been associated with a variety of neurological symptoms in children, including encephalitis and encephalopathy. However, B19 infection is not usually investigated in cases of symptomatic neurological diseases in newborns. In most reported cases, this virus was only sought after a negative finding of other pathogens commonly involved in meningoencephalitis.

In our series of 600 CSF samples from acute CNS infections B19 was identified in only a single sample (0.2%).

So far, not many case series of patients have analyzed the frequency of B19 among meningoencephalitis cases. Among them, this virus has been identified as a pathogen associated with undiagnosed meningoencephalitis, reaching a prevalence rate of up to 4.3%, a frequency higher than that obtained in our series. However, most of these series included a few numbers of patients with very heterogeneous clinical characteristics [14,27,28].

The present case report describes a patient of 38 days old with meningitis and encephalopathy associated with *Haemophilus influenzae* and Human Parvovirus B19. Her CSF also revealed pleocytosis with increased protein and lactate, and normal glucose levels. These findings are in agreement with prior descriptions of Parvovirus B19 CNS infection [14,29–31]. In blood tests, a decrease in hemoglobin with levels below 10 g/dL, and anisocytosis was observed, as reported in the literature [14,32,33]. The absence of a

skin rash in this patient during the clinical course is consistent with previous reports in patients with neurological manifestations caused by B19 infection [14].

Ten days after the onset of symptoms, imaging (CT) showed diffuse cerebral edema, diffuse subarachnoid hemorrhage, and multiple hypodense areas affecting the cortex and white matter in the frontal, temporal, parietal, and occipital lobes. All these alterations have been described among children acutely infected by either Parvovirus B19 or *Haemophilus influenza* [30,34,35]. Also, hydrocephalus and seizures are frequently described among the various neurological complications associated with both conditions [34–36].

The precise mechanism involved in the pathogenesis of B19 is still very controversial. Non-productive infection of B19 virus has been described in various tissues, including the CNS. It has been postulated that B19 infection stimulates cellular responses in the tissues against the virus which could culminate in a myriad of pathologies, such as meningoencephalitis [3,37]. B19 infection or the expression of viral proteins could modulate the immune response in the infected tissues by dysregulation of cytokines and production of autoantibodies, or by the cytotoxic effect of the B19 viral nonstructural protein damaging cerebrovascular endothelium [9,38–44].

It has also been postulated that when patients are under immunosuppression or during concomitant infection with other pathogens, the B19 viral load increases, which could evoke a more potent inflammatory disease. In this regard the immature immune response of the newborn could have also contributed to the severe clinical outcome observed in the present case. The overlapping infections of *Haemophilus influenzae* and Parvovirus B19, as described in the present case, could also be a determining factor involved in the severity of neurological manifestations observed [3,45].

Parvovirus B19 concomitant infection with other infectious agents have been described, such as the influenza virus, *Staphylococcus*, and mumps virus and many of such cases evolve with complications, serious sequelae or death [46–48]. As described above, it would be plausible to suppose that B19 infection or the expression of viral proteins could modulate the immune response in the CNS or other tissues and lead to different pathologies, such as meningoencephalitis, as reported in the present case [49].

To the best of our knowledge, this is the first case report of encephalopathy associated with *Haemophilus influenzae* and Human Parvovirus B19.

It is relevant to comment on the possibility of vertical parvovirus B19 transmission during pregnancy. Parvovirus B19 infects 1–5% of pregnant women, generally with normal pregnancy outcomes [3,50]. However, transmission of parvovirus B19 across the placenta can lead to fetal infection [50]. The risk of fetal complications depends largely on the gestational age at the time of maternal infection with parvovirus B19. However, fetal abnormalities associated with parvovirus B19 are rare. Whenever present they may result from injuries to different fetal organs including the brain, and as a result may also cause neurodevelopmental problems.

In the case reported here, according to information from the child's parents and medical records, pregnancy and delivery were uneventful, and the newborn developed normally until the 30th day of life, when *Haemophilus influenzae* meningitis was diagnosed. It is plausible to suppose that in case an asymptomatic parvovirus B19 infection occurred during pregnancy or just after delivery, the co-infection with *Haemophilus influenzae* may have triggered a more potent immune-mediated reaction which led to the observed neurological complications.

The Parvovirus B19 complete genome was recovered from the CSF and the genotypic showed, through phylogeny, sequences similar to B19 genotype 1. Genotype 1 is more prevalent in most parts of the world including Brazil [12,13,50–56], showing a close relationship with the Netherlands sequence (Genebank #JN211136). The other Brazilian sequences available in the reconstructions were relatively distant, suggesting a distant ancestor.

Assessing the nucleotide composition, we observed a variation of approximately 2% in sequences of the same genotype. This diversity is observed in single stranded DNA viruses due to their high mutation rate. According to our findings, there was no evidence

of possible modifications in the analyzed Parvovirus B19 genome that would justify its presence in the CSF or the severity of the clinical manifestation. So far, no genotypic pattern or possible modifications in B 19 genome has been specifically associated with the different clinical manifestations or severity of these manifestations in the literature. Our findings are in accordance with that. Further studies are necessary to evaluate the role of host's genetic factors or other viral factors which could be involved in the pathogenesis of Parvovirus B 19 in the CNS.

Based on our data we believe that metagenomics analysis of CSF in a patient with a severe neurological disorder may be an effective tool for investigating the presence of sequence diversity in a biological sample and could represent a novel protocol that may yield a more precise insight into viral evolution dynamics.

## 6. Conclusions

Based on the case analyzed here we conclude that 1—metagenomics analysis of a CSF sample is useful in the diagnosis of severe meningoencephalitis cases as a complement to traditional diagnostic protocols; 2—among patients with suspected CNS infection B19 could be involved more frequently than has been described so far, since it is not regularly tested for in cases of acute neurological manifestations.

**Author Contributions:** Conceptualization, N.E.F. and A.C.d.C.; investigation, N.E.F., L.H. and M.C.M.-C.; funding acquisition, E.G.K., M.C.M.-C. and H.R.G.; project administration, C.G.T.S. and A.C.S.d.O.; methodology, N.E.F., A.C.d.C., S.H.L., M.C.M.-C. and T.R.T.M.; sequencing analysis, N.E.F., A.C.d.C. and M.F.C.; writing—original draft preparation, N.E.F. and L.H.; data investigation, N.E.F. and H.G.O.P.; writing—review and editing, T.R.T.M., S.S.W., M.C.M.-C., D.d.M.V. and S.F.C. All authors have read and agreed to the published version of the manuscript.

**Funding:** This case report was funded FAPESP (Research Support Foundation of the State of São Paulo-SP—Brazil—Process Number 2017/10264-6) and Abbott Diagnostics, Abbott Park, IL, USA.

**Institutional Review Board Statement:** This study was approved by the local ethics committee (CONEP), protocol. No. CAAE: 67203417.0.0000.0068.

**Informed Consent Statement:** Informed consent was obtained from the patient's parents.

**Data Availability Statement:** All available data regarding this manuscript is presented in the present manuscript.

**Conflicts of Interest:** The authors declare no conflict of interest.

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
