# Peer review of "Encephalopathy Caused by Human Parvovirus B19 Genotype 1 Associated with Haemophilus influenzae Meningitis in a Newborn"

_cimb, doi:10.3390/cimb45090439_

Round 1

Reviewer 1 Report

Comments and Suggestions for Authors

Please round percentages to the same decimal place (L46, 50, 96, 190, 192, 250)

In MDPI style tittles of the chapters and subchapters shall be in the following font with the first letter of every word capitalized

L39-40 change to: classified in genus Erythroparvovirus of the Parvoviridae family

L46 it is not clear, seroprevalence from 40% to 87% in childhood or in general population?  

I suggest to shorten introduction, since the manuscript presents case report, but nor original paper and to write in concise way

L39-58 please shorten

L64-75 the description of the research should be clear concise.

L72-75 is conclusion/future perspectives not introduction

L77-91 has the results of this project/research been published in a scientific journal?

L99 here you refer to “the Hospital das Clinicas da Faculdade de Medicina da Universidade de Sao Paulo“?

L151, 181 exclude [24], this reference does not belong to MEGA11

L158 I do not see demographic data  (such as age, sex, geographic location, etc.) in Tables 1-2

L171 it is too few data for the separate Table, I suggest to describe data in the text. Furthermore, Table 3 looks like Figure rather than table

L 175 Change to Figure 1. Reword first sentence, it is not clear of what? of isolate/strain from the current study. “similarity/clustering” choose one.

L176-181 GenBank acc. no. have to be instead of 63660 Brazil

Phylogenetic analysis should be redone, phylogenetic tree has to be rooted, use the most related species to the analyzed group of taxa. The number of sequences must be displayed next to combined sequences into one cluster, for example, USA 2019 (n=2). Caption of Figure 1 must be improved. Now it is not clear what species/virus? Include that country and year of isolation is displayed next to GenBank accession numbers. In the figure 1, three genotypes are distinguished; however, there is no discussion about it in figure caption as well as in the text (L174-175). It is not clear how genotypes were distinguished; has it been done previously by other researchers? Genotypes represent monophyletic genetic lineages.

L192 check citing style through manuscript, it should be [14,26,27].

L191-192 what were the samples size? could you state that statistically significantly higher? Up to 4.3% and from? Please improve discussion on infection rates.

L210 why “B19 infection” is in boldface?

L214 abbreviation NS1 is not needed, since it is used only once.

L224 it should be “Staphylococcus

L223-226 there is no citation

L227-228 it is too short for separate paragraph, there is no discussion.

L237-243 no citation

L245 how genotypes are determined? Please specify. Are genotypes related to pathogenicity, clinical course, etc.?

L250 and what is the percentage genetic differences between three genotypes?

L244-255 where is no suggested explanation what traits distinguish different genotypes.

L182-255 the Discussion section must be improved, there is a lack of structuring of the Discussion by topics (for instance prevalence, clinical features, relationships with other pathogens, evolutionary aspects. Sections should be combined, in some sections there is a lack of citing and discussion.

L257-259 why some of the text is in boldface? Conclusions should be broadened explaining significance of the study.

L259 “involved” where? it is not clear.  

L262-263 I suggest including permissions related to ethical issues in the methodological part of the article.

References are not arranged in MDPI style. For instance, dots should be in some names of abbreviated journals; after journal name it should be dot; year of publishing must be in boldface; volume should be in italic; some of the article names are using sentence style and other using with the first letter of every word capitalized; etc.

Comments on the Quality of English Language

only minor chnages of language is needed 

Author Response

Reviewer 1

Comments and Suggestions for Authors

1-Please round percentages to the same decimal place (L46, 50, 96, 190, 192, 250)

Answer- We thank the Reviewer for this observation. We have made these changes, as suggested. Please refer to lines 48, 85 and 195.

2- In MDPI style tittles of the chapters and subchapters shall be in the following font with the first letter of every word capitalized

Answer- We thank the Reviewer for this observation. We have made these changes, as suggested. Please refer to lines 87,126,148,162 and 168.

L39-40 change to: classified in genus Erythroparvovirus of the Parvoviridae family

Answer- We have done as suggested. Please refer to lines 39 and 40

L46 it is not clear, seroprevalence from 40% to 87% in childhood or in general population? 

Answer- We thank the Reviewer for raising this point. We have rewritten the text to make this information clearer to the reader.  In fact, B19 infection seroprevalence ranges from 40% to 87% among the general population [6-8].

I suggest to shorten introduction, since the manuscript presents case report, but nor original paper and to write in concise way.

Answer- We thank the Reviewer for this observation and we have done as suggested.Please refer to Introduction, lines 38-65.

L39-58 please shorten

Answer- We thank the Reviewer for this observation, and we have done as suggested. Please refer to Introduction, lines 38-65.

L72-75 is conclusion/future perspectives not introduction.

Answer- We thank the Reviewer for raising this point. We agree with this observation, and we have rewritten the manuscript to include this observation in Discussion, as suggested. Please refer to the text lines 272-275.

L77-91 has the results of this project/research been published in a scientific journal?

Answer- The results of this project have not been published yet.

L99 here you refer to “the Hospital das Clinicas da Faculdade de Medicina da Universidade de Sao Paulo“?

Answer- We thank the Reviewer for this observation and indeed, we refer to “the Hospital das Clinicas da Faculdade de Medicina da Universidade de Sao Paulo”.We have modified the text to include this information, as suggested. Please refer to lines 89-90.

L151, 181 exclude [24], this reference does not belong to MEGA11.

Answer- We agree with the Reviewer. Indeed reference 24 does not apply here. We have modified the text.

L158 I do not see demographic data  (such as age, sex, geographic location, etc.) in Tables 1-2

Answer-We agree with the Reviewer and we have modified the Table, as suggested.

L171 it is too few data for the separate Table, I suggest to describe data in the text. Furthermore, Table 3 looks like Figure rather than table

Answer- We thank the Reviewer for raising this point and making this observation. We have included this information in the text. Please refer to lines 165-169.

L 175 Change to Figure 1. Reword first sentence, it is not clear of what? of isolate/strain from the current study. “similarity/clustering” choose one.

Answer-We have done as suggested. Please refer to the text.

L176-181 GenBank acc. no. have to be instead of 63660 Brazil

Phylogenetic analysis should be redone, phylogenetic tree has to be rooted, use the most related species to the analyzed group of taxa. The number of sequences must be displayed next to combined sequences into one cluster, for example, USA 2019 (n=2). Caption of Figure 1 must be improved. Now it is not clear what species/virus? Include that country and year of isolation is displayed next to GenBank accession numbers. In the figure 1, three genotypes are distinguished; however, there is no discussion about it in figure caption as well as in the text (L174-175). It is not clear how genotypes were distinguished; has it been done previously by other researchers? Genotypes represent monophyletic genetic lineages.

Answer-The phylogeny was rebuilt and rooted with genotypes 2/3 and we included more references from genotype 1. It was not possible to collapse the sequences into clusters by country/year because, as you may observe in the tree, the sequences from different countries are grouped in the same branch. We included in the sequence the name of the type of material and country of origin of the isolated viral genome sequence. The purpose of including the other genotypes in the analysis was only to prove that our sequence clusters with genotype 1. However, the B19 genotypes were established by Servant A, Laperche S, Lallemand F, Marinho V, De Saint Maur G, Meritet JF, Garbarg-Chenon A. Genetic diversity within human erythroviruses: identification of three genotypes. J. Virol. 2002, 76(18):9124-34.

L192 check citing style through manuscript, it should be [14,26,27].

Answer-We thank the reviewer for this observation. We have checked citing style through manuscript, as suggested.

L191-192 what were the samples size? could you state that statistically significantly higher? Up to 4.3% and from? Please improve discussion on infection rates.

Answer-We thank the reviewer for this observation. We have rewritten part of the Discussion.Please refer to lines 198-202.

L210 why “B19 infection” is in boldface?

Answer- We apologize for the mistake. And we have modified the text, as suggested.

L214 abbreviation NS1 is not needed, since it is used only once.

Answer-We thank the reviewer for this observation. We have excluded this abbreviation as suggested.

L224 it should be “Staphylococcus

Answer-We thank the reviewer for this observation. We have modified the text as suggested. Please refer to line 234.

L223-226 there is no citation.

Answer-We have included the references as suggested.

L227-228 it is too short for separate paragraph, there is no discussion.

Answer-We thank the reviewer for this observation. We have modified the text as suggested. Please refer to 235 and 238.

L237-243 no citation

Answer-We have included the references as suggested. Please refer to 235-238.

L245 how genotypes are determined? Please specify. Are genotypes related to pathogenicity, clinical course, etc.?

Answer-We thank the reviewer for raising this point. Please refer to lines 145-148 and 266-269.

L250 and what is the percentage genetic differences between three genotypes?

Answer-We have included this information in the text as suggested. Please refer to lines 145-148.

L244-255 where is no suggested explanation what traits distinguish different genotypes.

Answer-We thank the reviewer for raising this point. We have included this information in the manuscript. Please refer to lines 145-148 and 266-269.

L182-255 the Discussion section must be improved, there is a lack of structuring of the Discussion by topics (for instance prevalence, clinical features, relationships with other pathogens, evolutionary aspects. Sections should be combined, in some sections there is a lack of citing and discussion.

Answer- We have included additional information in Discussion and followed the Reviewer´s suggestions in order to improve it. Please refer to Discussion.

L257-259 why some of the text is in boldface? Conclusions should be broadened explaining significance of the study.

Answer- I am not sure why the text you received has some parts in boldface. Maybe the Editor could clarify this point.

Regarding the conclusions, we have made a few changes. Please refer to lines 277-281.

L259 “involved” where? it is not clear. 

Answer-We thank the reviewer for this observation. Indeed, we have modified the sentence to make the idea clearer to the reader. Please refer to lines 280.

L262-263 I suggest including permissions related to ethical issues in the methodological part of the article.

Answer- We thank the Reviewer for this observation, however, as described in the manuscript this study was approved by the local ethics committee (CONEP), protocol No. CAAE:  67203417.0.0000.0068, which includes permissions to all ethical issues involved with the different procedures of the study.

References are not arranged in MDPI style. For instance, dots should be in some names of abbreviated journals; after journal name it should be dot; year of publishing must be in boldface; volume should be in italic; some of the article names are using sentence style and other using with the first letter of every word capitalized; etc.

Answer- We thank the Reviewer for this observation, and we have modified the references, as suggested.

Reviewer 2 Report

Comments and Suggestions for Authors

This case report of bacterial meningitis in a neonate is interesting in microbiology. However, as a clinical report, the reviewers found significant problems with the argument of the case presentation. The reasons why the reviewers felt the argument was lacking are listed below, and the authors are encouraged to review each and every comment.

Major1: Even if parvovirus B19 was detected in the spinal fluid, there is no reason to attribute this case to infantile encephalopathy caused by parvovirus B19.

Major 2: The level of consciousness was not stated in stating the diagnosis of acute encephalopathy, and no electroencephalography was performed. The diagnosis of acute encephalopathy must be based on the definition of acute encephalopathy.

Minor as follows;

What is the type of hemophilus influenzae? Was this case of henophilus influenzae type B?

Was this patient not vaccinated against hemophilus influenzae?

Was there a low level of immunoglobulin?

Please present a CT brain image if possible.

In bacterial meningitis, the blood-brain barrier is disrupted due to inflammation of the meninges. In this state, parvovirus B19 can easily enter the CNS from the blood.

Have you checked the serum for parvovirus B19?

o you think that the route of transmission of parvovirus B19 is vertical transmission from the mother?

Why did the mother have a cesarean section?

Did the fetus have any findings suggestive of parvovirus infection after birth, such as fetal edema?

The name and dosage of the antimicrobial used to treat bacterial meningitis is not listed.

Was steroid or gamma globulin therapy administered?

If cytokines were measured, please add.

In neonatal infections, erythrocyte hyperplasia is often associated with chronic inflammation. Is this erythrocyte hyperplasia in this child caused by parvovirus?

Please describe the value of the erythrocytosis, as well as the white blood cell count and C-reactive protein (CRP).

Best Regards,

Dr Reviewer

Author Response

Reviewer 2

Major1: Even if parvovirus B19 was detected in the spinal fluid, there is no reason to attribute this case to infantile encephalopathy caused by parvovirus B19.

Answer- We thank the reviewer ‘s comments, however we respectfully disagree with it.

As extensively discussed in the manuscript the precise mechanism involved in the pathogenesis of B19 in the presented case is controversial (lines 218-232).

Many case reports have described parvovirus B19 infection in association with a variety of neurological manifestations, mainly in children. In a recent Systematic Review on parvovirus B19 infection and neurological manifestations, encephalitis and encephalopathy were the most common parvovirus B19 manifestations in the CNS and accounted for 38.8% of the total parvovirus B19- associated neurological manifestations, as mentioned in the manuscript. 

Also, non-productive infection of B19 virus has been described in various tissues, including the CNS. We agree with the Reviewer that the isolated presence of parvovirus B19 DNA in CSF could eventually be related to the persistence of parvovirus B19 DNA within the brain tissue of the child. However, taking into consideration the age of the child, the presence of the full genome of the virus and the presence of neurologic symptoms the attribution of causality to parvovirus may be likely and plausible, in our opinion.

Major 2: The level of consciousness was not stated in stating the diagnosis of acute encephalopathy, and no electroencephalography was performed. The diagnosis of acute encephalopathy must be based on the definition of acute encephalopathy.

Answer- We thank the Reviewer for raising this point.

Indeed, diagnosis of acute encephalopathy was done based on a clinical basis and was based on the definition of acute encephalopathy, as defined by current guidelines (1), as follows: “Acute encephalopathy (AE) denotes syndromes characterized by acute onset of severe and long-lasting disturbance of consciousness, which typically occur in previously healthy children. AE is the most severe complication of common infectious diseases, such as influenza and exanthem subitum, often leading to death or severe neurological disabilities”.

 As described in the manuscript (lines 88-126), a 38-day old girl was admitted to our hospital with a diagnosis of bacterial meningitis, seizures, and acute hydrocephalus on May 28th, 2019.She was immediately referred for additional analysis. In the hospital unit, meningitis due to Haemophilus Influenzae was diagnosed by a Polymerase Chain Reaction (PCR) assay and antibiotic therapy was initiated. Her symptoms worsened to persistent seizures requiring orotracheal intubation (OTI) and sedation. Computer tomography (CT) performed 2 days after symptom onset revealed supratentorial hydrocephalus, in addition to diffuse cerebral edema.

Taking into account all these considerations, we respectfully disagree with the Reviewer´s opinion that the diagnosis of acute encephalopathy could not be used in the case here reported.

Mizuguchi M, Ichiyama T, Imataka G, Okumura A, Goto T, Sakuma H, Takanashi JI, Murayama K, Yamagata T, Yamanouchi H, Fukuda T, Maegaki Y. Guidelines for the diagnosis and treatment of acute encephalopathy in childhood. Brain Dev. 2021 Jan;43(1):2-31.

Minor as follows.

What is the type of specific real-time PCR assay? Was this case of henophilus influenzae type B?

Answer-Unfortunately we do not have this information. A specific real-time PCR assay was performed for Hemophilus Influenzae, however information in the specific serotype was not available on the medical records.

Was this patient not vaccinated against hemophilus influenzae?

Answer-The patient was not vaccinated yet against Hemophilus Influenzae.

Was there a low level of immunoglobulin?

Answer-Unfortunately we were not able to obtain this information from medical records.

Please present a CT brain image if possible.

Answer- Unfortunately there is not an image to be presented in the manuscript at this time.

In bacterial meningitis, the blood-brain barrier is disrupted due to inflammation of the meninges. In this state, parvovirus B19 can easily enter the CNS from the blood.Have you checked the serum for parvovirus B19?

Answer-We thank the Reviewer for this comment, and we agree that in such a situation Parvovirus B 19 could have entered CNS from blood. However, we did not have access to blood samples to check for B 19 viremia or serologic markers.

Do you think that the route of transmission of parvovirus B19 is vertical transmission from the mother?

Answer- Indeed, It could be a possibility. As described in the manuscript (lines 241-248 )

Parvovirus B19 infects 1-5% of pregnant women. Transmission of parvovirus B19 across the placenta can lead to fetal infection [45]. The risk of fetal complications depends largely on the gestational age at the time of maternal infection with parvovirus B19.

In the case reported here, according to information from the child´s parents and medical records, pregnancy and delivery were uneventful, and the newborn developed normally until the 30th day of life, when Haemophilus influenzae meningitis was diagnosed.

It is plausible to suppose that in case an asymptomatic parvovirus B19 infection occurred during pregnancy or just after delivery, the co-infection with Haemophilus influenzae may have triggered a more potent immune-mediated reaction which led to the observed neurological complications.

Why did the mother have a cesarean section?

Answer-According to the medical records, there was no clinical reason for this type of decision. Apparently, it was a decision between the mother and the assistant physician.

Did the fetus have any findings suggestive of parvovirus infection after birth, such as fetal edema?

Answer-We thank the reviewer for this comment. However, as described in the manuscript               ( lines 249-255.) in the case reported here, according to information from the child´s parents and medical records, pregnancy and delivery were uneventful, and the newborn developed normally until the 30th day of life, when Haemophilus influenzae meningitis was diagnosed.

The name and dosage of the antimicrobial used to treat bacterial meningitis is not listed.

Answer- We thank the Reviewer for this important observation. The child received Ceftriaxone, 50mg/kg/dose (to a maximum of 2 grams) 24 hourly. We have included this information in the manuscript, lines 95-96.

Was steroid or gamma globulin therapy administered?

Answer-According to medical records steroids or gamma globulin were not prescribed.

If cytokines were measured, please add.

Answer-Unfortunately, as mentioned before, we did not have access to blood samples, and we did not measure cytokines.

In neonatal infections, erythrocyte hyperplasia is often associated with chronic inflammation. Is this erythrocyte hyperplasia in this child caused by parvovirus?

Answer-Unfortunately, we are not able to answer this question, at this point.

Please describe the value of erythrocytosis, as well as the white blood cell count and C-reactive protein (CRP).

Answer- We thank the Reviewer for these important observations. Evolution of blood analyses throughout case report including erythrocytes and white blood cell counts can be checked in Table 2. However, we do not have information on C-reactive protein at this time.

Reviewer 3 Report

Comments and Suggestions for Authors

Review is includede in the attached file.

Author Response

Reviewer 3

Comments and Suggestions for Authors

Reviewers’ comment

Answer

Comment on round 2

1.The B19 virus was identified in

a sample collected on June 6.

Why did the authors not confirm

the virus's presence in the

other samples?

Regarding the analyses of

additional samples,

unfortunately, we were not

able to analyze other samples

from the newborn, because

there was not enough CSF

volume from these samples,

collected at other dates, to

be submitted to complementary

analyses

Considering that this paper is a

case report, as the authors

claim, it is acceptable that

quantitative parvovirus B19 PCR

testing was not performed on all

samples.

However, as shown in Table 2, the authors collected

cerebrospinal fluid on nine

occasions and conducted a 13-item

clinical test. It is considered

that multiple rounds of

qualitative PCR were possible. By

all means, I would like you to

add the qualitative PCR test

results on the revised

manuscript

1-Answer to Comment – We thank the Reviewer for this comment.

However, it is important to mention that all information on Tables 1 and 2 , was obtained from the electronic laboratory system of our hospital.  All tests were performed at the Central Laboratory, and we were not able to have access to this material.

Reviewers’ comment 2

Answer

Comment on round 2

2-It seems that the B19 virus was

a consequence of Haemophilus

influenza infection and this

bacteria was responsible for

the neurological problems.

As extensively discussed in

the manuscript the precise

mechanism involved in the

pathogenesis of B19 in this

specific case is

controversial.

Determination of the exact

mechanism is difficult, as the

authors argue. Instead, the

discussion in this paper is based

on the whole genome information

of the detected Parvovirus B19.

Therefore, if the GenBank

accession number of the whole

genome information determined

from the author and the analysis

based on the whole genome

information were performed, even

if the relationship between

Haemophilus influenza infection

and onset was not determined, I

think is worthy of publication in

this journal.

Answer to Comment 2-– We thank the Reviewer for this comment.

Indeed, as described in the manuscript (lines 165-169), the Parvovirus B19 complete genome was recovered from the CSF sample by a metagenomic approach. This complete genome sequence (sample ID 63660- Mapped reads 546.585, genome length 5427, coverage 34.214, % B19 genome 100) has been deposited in GenBank under the accession number OR200802.

Reviewers’ comment 3

Answer

Comment on round 2

The conclusion should be

rephrased; the metagenomic

analysis of CSF samples has been done previously (PMID:

31189036; PMID: 29305150) and other reports have described a higher prevalence of this virus

(4.3%) contrary to your results (0.17%).

We have rephrased the text as suggested, to highlight this important point. Please

check lines 275-277.

The line numbers specified by the author refer to the References, not to the edited text.

I was not able to confirm any changes in the

text referring to previous studies as suggested by Reviewer1.

Answer to Comment 3 – I am sorry to know there was some kind of trouble when I submitted a new version of the manuscript. Again, I confirm I have rephrased the text as suggested, as follows (lines 277-282  ) :

Conclusions

Based on the case analyzed here we conclude that: 1- Metagenomics analysis of a CSF sample is useful in the diagnosis of severe meningoencephalitis cases as a complement to traditional diagnostic protocols; 2- Among patients with suspected CNS infection B19 could be involved more frequently than has been described so far, since it is not regularly tested for in cases of acute neurological manifestations.

Reviewers’ comment 4

Answer

Comment on round 2

Please be consistent when

describing the age; use days or

months

Please be consistent when

describing the age; use days

or months

I have verified that the authors

have made the appropriate changes

Answer to Comment 4– No additional comments.

Reviewers’ comment 5

Answer

Comment on round 2

The study was also conducted in

a single country, limiting its

generalizability to other

populations. Additionally, the study's methodology could be more explicitly stated, and the

results could be better

presented.

To the best of our knowledge, this is the first case report of encephalopathy associated

with Haemophilus Influenzae

and Human Parvovirus

B19.Therefore, we decided to present the case as a case

report.

Regarding the methodology

used, we added a few more

information on bioinformatic

analysis, as suggested.

Please check lines 152-154.

Regarding the presentation of results, we have done as

suggested, and included a fewmodifications in Table 1, in order to improve the text.

Please check Tables 1 and 2

and other modification on lines 164-176.

I cannot read the modifications on

the lines which stated in the

response of authors.

Please re-check the lines of the modifications.

Answer to Comment 5–We have rewritten part of Methods and Results.Please refer to the manuscript, lines 127-187.

Figure 1: Genbank-wide information on parvovirus B19 is relatively scarce, and the phylogenetic tree

in Figure 1 does not provide geographical information on transmission. On the other hand, GenBank has a lot of information about a gene , so please create a phylogenetic tree based on the homology of the gene instead of the whole genome.

Answer- We thank the Reviewer for this important observation. And, following the Reviewer’s suggestions we performed a new phylogenetic analysis, by including more sequences with a new prototype from different geographic regions. As a result, we did obtain a new tree , which is shown below in Figure 1 , as follows:

Answer:

Figure 1: Prototype from different geographic regions in different tissues

As can be observed above, we did not observe any change in the topology of the tree. Indeed, we analyzed other distinct genes individually by phylogenetic reconstruction of genotype 1, and their topology remained the same as the full genome analysis in the tree presented in the original version of the manuscript. Under these new conditions, the group of sequences in which our sample is included remained the same.

Additionally, we received the following comment from another Reviewer:

“Phylogenetic analysis should be redone, phylogenetic tree has to be rooted, use the most related species to the analysed group of taxa. The number of sequences must be displayed next to combined sequences into one cluster, for example, USA 2019 (n=2). Include that country and year of isolation is displayed next to GenBank accession numbers”.

Following his/her suggestions, the phylogeny was rebuilt and rooted with genotypes 2/3 and we included more references from genotype 1.

However, it was not possible to collapse the sequences into clusters by country/year because, as may observed in the tree, the sequences from different countries are grouped in the same branch.

As described in the manuscript (170-187)  phylogenetic analyses demonstrated similarity with genotype 1 B19 sequences compared to GenBank sequences.

The tree was constructed using the maximum likelihood approach, and branching support was estimated using an ultrafast bootstrap test with 1000 replications using the IQ-Tree tool (http://iqtree.cibiv.univie.ac.at/). Tree was visualized and edited using Figtree version 1.4.2 (http://tree.bio.ed.ac.uk/software/figtree).

6- Page 5 line 170: “GenBank under the accession number pending” -> Please obtain Genbank's accession number and describe it before resubmission.

Answer- We have done as suggested. Please refer to 168-169.

Round 2

Reviewer 1 Report

Comments and Suggestions for Authors

The manuscript is greatly improved and could be accepted after minor changes.

L48-50 Please change to: “Most cases of B19 infection are asymptomatic, occur more frequently in childhood and are not uniform across the population because seroprevalence ranges from 40% to 87%...”

L69 change 1 to “one”

L158-159 maybe better would be “The classification of the Parvovirus B19 genotypes (G1, G2 and G3) was based on the genetic variability of the nucleotide sequences with a variation between 5% and 13%”

L204-205 which analysis method was used for multiple sequence alignment, CLUSTALW or MUSCLE?

Figure 1. when writing percentages, replace commas with dots. For example 91.89 should be. Please round percentages to one decimal place. The topology and structure of phylogenetic tree is good, however, the quality of the figure is not well. Fonts are too small. I suggest to modify figure using CorelDRAW or other image editing programs.

Comments on the Quality of English Language

The manuscript is greatly improved and could be accepted after minor changes. However, the resolutions power of the phylogenetic tree should be better for the publishing.

Author Response

Reviewer 1

L48-50 Please change to “Most cases of B19 infection are asymptomatic, occur more frequently in childhood and are not uniform across the population because seroprevalence ranges from 40% to 87%...”

Answer: He have done as suggested.  Please refer to lines 43-46.

L69 change 1 to “one”.

Answer: He have done as suggested.  Please refer to line 181.

L158-159 maybe better would be “The classification of the Parvovirus B19 genotypes (G1, G2 and G3) was based on the genetic variability of the nucleotide sequences with a variation between 5% and 13%”.

Answer: He have done as suggested.  Please refer to lines 145-147.

L204-205 which analysis method was used for multiple sequence alignment, CLUSTALW or MUSCLE?

Answer: The sequences were aligned using the MAFFT tool. We have included this information in the manuscript as suggested. Please refer to line 170 and 361-361 (reference 25).

Figure 1. when writing percentages, replace commas with dots. For example, 91.89 should be. Please round percentages to one decimal place. The topology and structure of phylogenetic tree is good; however, the quality of the figure is not well. Fonts are too small. I suggest to modify figure using CorelDRAW or other image editing programs.

Answer: Percentages were rounded, and the commas were replaced with dots, as suggested.

The figure was edited again by other image editing programs, in order to make the numbers appear as clear as possible. Please refer to Figure 1 and to Supplementary Material (Figure 1 S).

Reviewer 3 Report

Comments and Suggestions for Authors

I appreciate the author's careful explanation and appropriate response. I recommend that this paper be published in its present form.

Author Response

Reviewer 3 did not have any additional comments.